# B-Cell Epitopes-Based Chimeric Protein from SARS-CoV-2 N and S Proteins Is Recognized by Specific Antibodies in Serum and Urine Samples from Patients

**DOI:** 10.3390/v15091877

**Published:** 2023-09-05

**Authors:** Fernanda F. Ramos, Isabela A. G. Pereira, Mariana M. Cardoso, Raquel S. Bandeira, Daniela P. Lage, Rahisa Scussel, Rafaela S. Anastacio, Victor G. Freire, Marina F. N. Melo, Joao A. Oliveira-da-Silva, Vivian T. Martins, Grasiele S. V. Tavares, Danniele L. Vale, Camila S. Freitas, Ana Thereza Chaves, Júlia F. M. Caporali, Paula F. Vassallo, Cecilia G. Ravetti, Vandack Nobre, Flavio G. Fonseca, Myron Christodoulides, Ricardo A. Machado-de-Ávila, Eduardo A. F. Coelho, Fernanda Ludolf

**Affiliations:** 1Programa de Pós-Graduação em Ciências da Saúde: Infectologia e Medicina Tropical, Faculdade de Medicina, Universidade Federal de Minas Gerais, Belo Horizonte 30130-100, Minas Gerais, Brazil; fe.fonsecaramos@gmail.com (F.F.R.); amorim.gpereira@gmail.com (I.A.G.P.); marianademeloc@gmail.com (M.M.C.); eduardoferrazcoelho@yahoo.com.br (E.A.F.C.); 2Programa de Pós-Graduação em Ciências da Saúde, Universidade do Extremo Sul Catarinense, Criciúma 88806-000, Santa Catarina, Brazilr_andrez@yahoo.com.br (R.A.M.-d.-Á.); 3Departamento de Clínica Médica, Faculdade de Medicina, Universidade Federal de Minas Gerais, Belo Horizonte 30130-100, Minas Gerais, Brazil; 4Centro de Tecnologia de Vacinas (CT Vacinas), BH-Tec, UFMG, Belo Horizonte 31270-901, Minas Gerais, Brazil; 5Laboratório de Virologia Molecular e Aplicada, Departamento de Microbiologia, Instituto de Ciências Biológicas, Universidade Federal de Minas Gerais, Belo Horizonte 31270-901, Minas Gerais, Brazil; 6Neisseria Research Group, Molecular Microbiology, School of Clinical and Experimental Sciences, University of Southampton Faculty of Medicine, Southampton General Hospital, Southampton SO16 6YD, UK; 7Departamento de Patologia Clínica, COLTEC, Universidade Federal de Minas Gerais, Belo Horizonte 31270-901, Minas Gerais, Brazil; 8Programa de Pós-Graduação em Ciências da Saúde, Faculdade Ciências Médicas de Minas Gerais, Belo Horizonte 30130-110, Minas Gerais, Brazil

**Keywords:** B-cell epitopes, chimeric protein, diagnosis, urine, serum, SARS-CoV-2

## Abstract

The impact of the COVID-19 pandemic caused by the SARS-CoV-2 virus underscored the crucial role of laboratorial tests as a strategy to control the disease, mainly to indicate the presence of specific antibodies in human samples from infected patients. Therefore, suitable recombinant antigens are relevant for the development of reliable tests, and so far, single recombinant proteins have been used. In this context, B-cell epitopes-based chimeric proteins can be an alternative to obtain tests with high accuracy through easier and cheaper production. The present study used bioinformatics tools to select specific B-cell epitopes from the spike (S) and the nucleocapsid (N) proteins from the SARS-CoV-2 virus, aiming to produce a novel recombinant chimeric antigen (N4S11-SC2). Eleven S and four N-derived B-cell epitopes were predicted and used to construct the N4S11-SC2 protein, which was analyzed in a recombinant format against serum and urine samples, by means of an in house-ELISA. Specific antibodies were detected in the serum and urine samples of COVID-19 patients, which were previously confirmed by qRT-PCR. Results showed that N4S11-SC2 presented 83.7% sensitivity and 100% specificity when using sera samples, and 91.1% sensitivity and 100% specificity using urine samples. Comparable findings were achieved with paired urine samples when compared to N and S recombinant proteins expressed in prokaryotic systems. However, better results were reached for N4S11-SC2 in comparison to the S recombinant protein when using paired serum samples. Anti-N4S11-SC2 antibodies were not clearly identified in Janssen Ad26.COV2.S COVID-19-vaccinated subjects, using serum or paired urine samples. In conclusion, this study presents a new chimeric recombinant antigen expressed in a prokaryotic system that could be considered as an alternative diagnostic marker for the SARS-CoV-2 infection, with the potential benefits to be used on serum or urine from infected patients.

## 1. Introduction

Considering the impact of the COVID-19 pandemic caused by the SARS-CoV-2 virus, scientists around the world have continued to look for new solutions to curb virus transmission and prevent a new wave of this life-threatening disease. In vitro diagnostic assays have proven to be a critical part of the complete strategy to control the COVID-19 pandemic. While serological tests are not currently applicable to diagnose an acute infection on their own, they can indicate the presence of antibodies generated from a previous infection and/or vaccination [1]. In addition to the importance of serological tests at the population level to support surveillance studies, they are currently relevant for patients seeking medical care with late diagnosis, those with co-morbidities and complications of the disease presenting persistent symptoms caused by ‘long COVID’ [2,3,4].

The SARS-CoV-2 virus has mutated over time, resulting in genetic variation in the population of circulating viral strains. Such mutations have impacted the diagnostic performance of distinct molecular and serological tests leading to reduced accuracy. Therefore, it is relevant to search and obtain alternative diagnostic options aiming to improve the quality of the diagnosis of infection, especially among the vulnerable populations presenting comorbidities [5].

Selecting SARS-CoV-2 recombinant proteins is essential for developing a reliable serological test. SARS-CoV-2 is an envelope virus with a positive-sense, single-stranded RNA genome, comprising several non-structural proteins and four structural proteins: Nucleocapsid (N), Membrane (M), Spike (S) and Envelop (E). S protein is a surface-exposed protein which plays an important role during the viral infection by binding to the angiotensin-converting enzyme 2 (ACE-2) receptors of the host cells. N protein is an RNA-binding protein crucial for the replication and transcription of SARS-CoV-2, being highly expressed during infection [6,7]. Among the four structural proteins of SARS-CoV-2, N and S proteins are the most immunogenic and, therefore, the most used in serological tests [8,9]. The SARS-CoV-2 N protein can be efficiently expressed in prokaryotic system(s) maintaining good immunoreactivity; however, the S protein has been preferentially expressed in eukaryotic systems, which often generate post-translational modifications of this antigen [10].

Laboratorial tests have used serum samples to detect the disease. Although considered less invasive than swabs, the sample’s collection presents a significant rate of complication, since it can be unpleasant and requires a trained phlebotomist. By contrast, the collection of urine to detect specific antibodies is less costly and samples are easy to store and thus could be convenient for clinical and epidemiological studies [11]. Urine-based tests to detect antibodies have been recently suggested as a non-invasive, simple and safe alternative to detect anti-SARS-CoV-2 N and S antibodies. In these studies, a urine-based ELISA targeting to identify anti-N protein antibodies showed sensitivity and specificity of 94.0% and 100%, respectively. Meanwhile, the same assay using the S protein purified from a prokaryotic system showed sensitivity and specificity of 89.0% and 97.0%, respectively. Interestingly, when sera samples collected from the same patients were tested against the prokaryotic S antigen, results showed sensitivity of 40.0% and specificity of 98.0%, highlighting the difficulty to obtain antigens with high accuracy derived of prokaryotic systems of purification [11,12].

Despite the biological sample evaluated, diagnostic antigens are required to be used in sensitive and specific laboratorial tests. In this context, B-cell epitope-based chimeric proteins have been proposed as advantageous in comparison to individual proteins, since they show higher antigenicity against diverse biological samples [13,14]. In addition, bioinformatics tools used to predict B-cell epitopes offer advantages in terms of speed and biosafety, being unbiased by specific peptide selection [15]. Bioinformatics and/or microarray analyses have been applied to identify B-cell epitopes derived from SARS-CoV-2, with corresponding peptides demonstrating a good diagnostic performance [10,14,16,17,18]. Javadi Mamaghani et al. [13] designed a multi-epitope SARS-CoV-2 protein and the antigen showed good serodiagnostic efficacy. B-cell epitopes of SARS-CoV-2 identified by SPOT synthesis analysis were incorporated as a chimeric protein, and the antigen showed also diagnostic potential against the infection [19].

In this study, we apply bioinformatic tools to design a new chimeric protein containing specific B-cell epitopes predicted by the amino acid sequences from S and N proteins of SARS-CoV-2 virus, with the purpose to develop an alternative antigen for the detection of COVID-19 cases, which will be easy to produce and presents better cost-effective conditions. The results obtained in the in-house ELISA experiments indicated that the chimeric protein, which was expressed in a prokaryotic system, could be effectively employed as a target to reach a high-performance diagnostic test, by using both serum and urine samples from patients.

## 2. Material and Methods

### 2.1. Research Subjects

This study was approved by the Human Research Ethics Committee from Federal University of Minas Gerais (UFMG, Belo Horizonte, Brazil) with protocol number CAAE 30,437,020.9.0000.5149. All included participants were male or female adults who signed an informed consent form. Patients (*n* = 79) with clinical symptoms and seeking hospital assistance were assessed by the attending physician and included in this study after confirmation of SARS-CoV-2 infection by positive qRT-PCR.

### 2.2. Biological Samples

Paired urine and serum samples (*n* = 135) from hospitalized patients were collected on the first day of inclusion and whenever possible, on days 1, 3, 7 and 14 after recruitment; thus, varying the corresponding day Post-Symptom Onset (PSO) for each patient. These samples were collected before COVID-19 vaccination began in Brazil. Samples collected before 2019 were considered truly negative and called “pre-COVID-19 negative” (*n* = 10 urine and *n* = 14 sera). Samples from subjects who had maintained a rigorous quarantine and did not show any symptoms, were considered theoretically negative and called “post-COVID-19 negative” (*n* = 11 urine and *n* = 6 sera). Samples from Janssen Ad26.COV2.S COVID-19-vaccinated subjects (*n* = 40) were included in this study, and they were collected after COVID-19 vaccination began in Brazil. Urine and serum samples were collected and stored as described by Ludolf et al. [12]. Briefly, urines were diluted in 0.1% (*w*/*v*) sodium azide and stored at 4 °C, while sera samples were stored at −20 °C, until use.

### 2.3. Prediction of B-Cell Epitopes and Construction of Chimeric Protein

The amino acid sequences of surface glycoprotein (YP_009724390.1) and nucleocapsid phosphoprotein (YP_009724397.2) proteins [Severe acute respiratory syndrome coronavirus 2] were obtained from the Genbank. The IEDB server v2.26 (www.iedb.org, accessed on 16 August 2023) was used to identify the most accessible amino acids in the primary structures using the B-Cell epitope prediction tool, through the parameter “Antigen Sequence Properties”. The method Emini Surface Accessibility Prediction was chosen with different window sizes of 14, 12 and 10. All the amino acids with the threshold value above 1.0 were considered. Next, the ABCpred server (www.imtech.res.in/raghava/abcpred/, accessed on 16 August 2023) was used to predict the B-cell epitopes with window sizes of 14, 12 and 10. All the sequences with the threshold value above 0.85 were considered. Overlapping regions were assembled using Clustal Ômega tool (https://www.ebi.ac.uk/Tools/msa/clustalo/, accessed on 16 August 2023), comparing all the parameters with each other and with the existing literature. The most frequent regions were considered as the final predicted epitopes. After assessment, the possible sequences were observed in the 3D structure of the proteins to analyze their position and distribution (Figure 1A,B). At this point, the spatial configuration of the proteins was analyzed to observe if the predicted sequences were positioned in a way to enable and/or facilitate recognition. To do so, the structures were obtained from Protein Data Bank (PDB) (https://www.rcsb.org/, accessed on 16 August 2023), and epitopes were selected and identified using the software SwissPDB-viewer v4.10. For the surface glycoprotein (YP_009724390.1) and the nucleocapsid phosphoprotein (YP_009724397.2), the PDB structures “6xr8A” and “8FD5” were used, respectively. Eleven Spike and four Nucleocapsid B-cell epitopes were selected, and their amino acid sequences were joined by –GPGPG- linker peptides aiming to provide flexibility and to avoid spatial overlap [20]. The arrangement of the peptides in the chimeric protein was distributed in a position as similar as possible to the original proteins, preserving the order in which they appear. The Protparam tool from Expasy (https://web.expasy.org/protparam/, accessed on 16 August 2023) was used for the physical–chemical characterization of the multi-epitope chimeric protein, which was called N4S11-SC2 (Figure 1C). In this step, the new protein was analyzed again by ABCPred to confirm if the peptides would still be suggested as antigenic regions.

### 2.4. Production of Recombinant N4S11-SC2 Protein

The chimeric protein-codifying gene sequence was commercially synthetized as having 1107 bp and it was cloned into prokaryotic expression vector pET28a-TEV (Genscript^®^, Piscataway, NJ, USA). The construct was transformed into *Escherichia coli* BL-21 strain cells, and the protein was expressed by addition of 0.2 mM isopropyl-β-d-thiogalactopyranoside (IPTG; Promega, Madison, WI, USA) for 2 h at 37 °C in 1 L of LB medium (with yield of 36 mg/L of culture). Next, bacteria were centrifuged at 3000× *g* for 10 min at 4 °C and suspended in lysis buffer (20 mM Tris-HCL, pH 8.0; 0.5 M NaCl; 5 mM imidazole; 8 M urea; 1 mM β-mercaptoetanol), followed by six cycles of ultrasonication for 30 s each (at 90 Hz). Cellular debris was removed after 5000× *g* centrifugation for 15 min at 4 °C and the supernatant was collected. The N4S11-SC2 protein was purified on a HisTrap HP affinity column (GE Healthcare Life Sciences, Chicago, IL, USA) connected to an AKTA system. Inclusion bodies were solubilized in 2 M urea buffer. A 12.5% (*w*/*v*) SDS-PAGE was done to evaluate the purity of the recombinant protein. A Page Ruler broad range unstained protein ladder (Thermo Fisher Scientific Baltics, Vilnius, Lithuania) was used.

### 2.5. Immunoblottings

Immunoblottings were performed using the purified N4S11-SC2 protein, which was applied (2 μg) to 12.5% SDS-PAGEs and blotted onto a nitrocellulose membrane (0.2 μm pore size, Sigma-Aldrich/Merck, Darmstadt, Germany). Next, membrane was blocked with a solution of phosphate buffered saline pH 7.4 (PBS) plus 0.05% (*v*/*v*) Tween 20 (PBS-T) and 1% (*w*/*v*) bovine serum albumin (BSA; Sigma-Aldrich/Merck, Darmstadt, Germany), for 16 h at 37 °C, followed by incubation with 6-HIS tag antibody (1 mg/mL, MA1-21315; Invitrogen; Rockford, IL, USA). Infected (*n* = 9) and negative pre-pandemic sera (*n* = 9) pools were added for 1 h at 37 °C, with samples diluted 1:100 in in PBS-T and anti-HIS antibody diluted 1:3000 in PBS-T. Next, membranes were washed with PBS-T and an anti-human goat IgG peroxidase conjugated antibody (A18811; Invitrogen; Carlsbad, CA, USA) was added in the plates (diluted 1:15,000 in PBS-T), with a new incubation occurring for 1 h at 37 °C. After, reactions were developed using a solution composed by chloronaphtol, diaminobenzidine and H_2_O_2_ for 30 min, and stopped by adding distilled water. A Page Ruler pre-stained protein ladder (ThermoFisher Scientific Baltics, Vilnius, Lithuania) was used.

### 2.6. ELISA

ELISA experiments were conducted according to Ludolf et al., with few modifications [21]. Briefly, previous titration curves were performed to determine the most appropriate concentration of antigens, antibodies/sample dilution and time of incubations to be used in the assays. High-binding ELISA plates (Nunc MaxiSorp™ flat-bottom; Invitrogen by Thermo Fisher Scientific, Carlsbad, CA, USA) were coated with 400 ng/well of N4S11-SC2 diluted in carbonate buffer pH 9.6 for 18 h at 4 °C. Wells were then blocked with a solution of PBS-T and 1% (*w*/*v*) BSA for 2 h at 37 °C. Then, 100 µL/well of urine (undiluted) or serum (diluted 1:100 in PBS-T) samples were added and the incubation occurred for 1 h or 30 min for urine or serum samples, respectively, at 37 °C, after which they were again washed. Peroxidase-conjugated anti-human IgG antibody (A18811, 1/10,000 dilution in PBS-T; Invitrogen, Carlsbad, CA, USA) was added to the wells and plates were incubated for 1 h or 30 min at 37 °C, for urine or serum samples, respectively. Next, wells were washed and reactions were developed by addition of TMB (3,3’,5,5-tetramethylbenzidine) for 15 min in the dark. Reactions were stopped by adding 0.5 M H_2_SO_4_ and the optical density (OD) values were read on a microplate spectrophotometer (Multiskan Go, ThermoFisher, Finland), at 450 nm. The cut-off values were determined as the mean plus 2.5 and 2.3 times the standard deviation of negative samples for urine and serum assay, respectively. The index (I) value for each sample was calculated using the equation I = (OD λ450 nm)/(cut-off). The index value was classified as positive above 1.1, indeterminate between 0.8 and 1.1 and negative below 0.8.

The prokaryotic recombinant proteins, N and S (Prok2-S1) were validated by Ludolf et al. [12] and Ramos et al. [11]. These antigens were used as controls to compare the performance of chimeric protein by using serum and urine samples (*n* = 72 each), which were obtained after eight days PSO.

### 2.7. Statistical Analysis

Data were analyzed using the GraphPad Prism^TM^ program (version 8.0 for Windows; La Jolla, CA, USA). Value distributions [mean (M) ± standard deviation (SD), as indicated] were obtained for continuous variables, while categorical ones were evaluated as proportions. Receiver Operator Characteristic (ROC) curves were constructed with the OD values of positive (SARS-CoV-2 infection) versus negative (pre-COVID-19 and post-COVID negative) samples. Diagnostic performance was evaluated by estimation of sensitivity (Se), specificity (Sp), Area Under the Curve (AUC) and Youden index (J). Confidence intervals (CI) were defined at the 95% confidence level (95% CI). A paired t-test was used to compare the distinct groups and *p* < 0.05 values were considered significant. Positive and Negative Predictive Values (PPV and NPV, respectively) were calculated based on the index value, excluding the indeterminate value samples, and using the equation NPV = true negative/false negative + true negative and PPV = true positive/false positive + true positive.

## 3. Results

### 3.1. Construction of the Chimeric Protein

The amino acid sequences of surface glycoprotein (YP_009724390.1) and nucleocapsid phosphoprotein (YP_009724397.2) proteins were evaluated by bioinformatics and the results are shown (Figure 1A). Eleven Spike and four Nucleocapsid B-cell epitopes were selected (Appendix A) and their amino acid sequences were used to construct the chimera-codifying gene in the pET-TEV vector. The amino acid sequences were: TRTQLPPAYTNSFTRGVYY, VYYHKNNKSWMESEFRVY, TRFASVYAWNRKRISN, QIAPGQTGKIADYNYKLP, SKVGGNYNYLYRLFRKSNLKPFERDIST, YFPLQSYGFQPTNGVGY, TESNKKFLPFQQFGRDIADTTDAVRDP, HADQLTPTWRVYSTGSN, SYQTQTNSPRRARS, ILPDPSKPSKRSFIEDLLFNKV and FKEELDKYFKNHTSPDVDLGD for the S protein and TGSNQNGERSGARSKQRRPQG, TNSSPDDQIGYYRRATRR, RSSSRSRNSSRNSTPGS and GQTVTKKSAAEASKKPRQKRT for the N protein. All epitopes were joined by –GPGPG- linker peptides aiming to provide flexibility and to avoid spatial overlap. The recombinant protein N4AS11-SC2 presents an N-terminal tag (MGHHHHHHENLYFQGHMAS) containing the initial methionine and HIS tag from the pET-TEV vector (Figure 1C).

### 3.2. Characterization of Recombinant N4S11-SC2 Protein

A physical–chemical characterization of the N4S11-SC2 protein was performed and results showed a molecular weight of 41.3 KDa, an isoelectric point of 10.17, an estimated half-life of >10 h in *E. coli* and an instability index (II) of 38.08, suggesting that the protein is stable. SDS-PAGE showed a band of ~41.0 KDa, as expected (Figure 2A). The expression of the N4S11-SC2 protein was confirmed by an immunoblotting assay using an anti-HIS monoclonal antibody. The immunoblottings indicated a stronger positive recognition of the N4S11-SC2 protein to the COVID-19 patient’s pools as compared to pre-pandemic negative serum pools (Figure 2B).

### 3.3. Diagnostic Evaluation of Chimeric N4S11-SC2 Protein

The immunodiagnostic efficacy of the N4S11-SC2 protein to detect COVID-19 cases was evaluated using an in-house serum- and urine-based ELISA. Paired samples from qRT-PCR-positive patients were used, as well as unpaired negative samples from pre-COVID-19 and post-COVID-19 individuals. Those samples were collected before vaccination had started in Brazil. The ELISA testing urine samples resulted in sensitivity and specificity values of 91.1% and 100%, respectively, while assays performed using serum samples showed sensitivity and specificity values of 83.7% and 100%, respectively (Table 1).

ROC curves were constructed to assess the accuracy of the assays and the results indicated that the urine-based ELISA had a slightly better accuracy with an AUC of 0.9841, as compared to the serum-based ELISA that presented an AUC of 0.9722 (Figure 3A). To standardize and compare the results, individual index values (I) were calculated from OD values and the data of the urine-based ELISA demonstrated that out of 135 samples collected from 79 patients who tested positive for qRT-PCR on different days post-symptom onset (PSO), 118 samples exhibited a positive reaction with the N4S11-SC2 protein, as indicated by a positive index value above 1.1. Additionally, eight samples were classified as “indeterminate” with index values ranging from 0.8 to 1.1, while nine samples showed a negative index value below 0.8. Among the 21 negative control urine samples, two of them showed an indeterminate index value, whereas none exhibited a positive index value (Figure 3B).

Simultaneously, 108 out of 135 serum samples collected from 79 patients who tested positive for qRT-PCR on different days post-symptom onset (PSO) exhibited a positive reaction to the N4S11-SC2 protein, with positive index values exceeding 1.1. Additionally, 16 samples were classified as “indeterminate”, with index values ranging from 0.8 to 1.1, and 11 samples presented a negative index value below 0.8 Among 20 negative control serum samples, only three showed “indeterminate” index values, and none of them displayed positive index values (Figure 3B). PPV and NPV were calculated based on the index values, which excluded the “indeterminate” index value samples, and the NPV and PPV results were 0.679 and 1.000 for urine, respectively, and 0.607 and 1.000 for serum, respectively. Samples (*n* = 40) from Janssen Ad26.COV2.S COVID-19-vaccinated subjects included in the study indicated that two serum and eleven urine samples exhibited reactions against the N4S11-SC2 protein, indicating a positive index value exceeding 1.1. Additionally, six serum and three urine samples were classified as indeterminate, with index values ranging from 0.8 to 1.1, and 32 serum and 26 urine samples showed negative index values below 0.8 (Figure 3B). Comparable findings were achieved using the same pool of urine samples for the N4S11-SC2 protein when compared to the N and S recombinant proteins expressed in the prokaryotic system. However, higher sensitivity was observed for the N4S11-SC2 protein, as compared to the S recombinant protein using the same pooled serum samples (Figure 4). 

## 4. Discussion

In vitro diagnostic assays have shown importance in helping control the COVID-19 pandemic. Serological tests may indicate the presence of antibodies from a previous infection or vaccination, and researchers have remained continuous in the search for new diagnostic solutions to prevent an eventual new wave of this disease in the world. Therefore, the development of more suitable and cost-effective diagnostic antigens is still desirable. Given the constant emergence of new variants, having alternative diagnostic options in place also becomes of paramount importance. In the present study, we constructed a chimera protein called N4S11-SC2 by combining linear B-cell epitopes predicted from amino acid sequences of the Nucleocapsid and Spike proteins from SARS-CoV-2. N4S11-SC2 was expressed in a prokaryotic system and used in an in-house ELISA platform against serum and urine samples from COVID-19 patients diagnosed by RT-PCR. The major findings from the study were that N4S11-SC2 showed relevant sensitivity and specificity values for the diagnosis of SAR-CoV-2, by using the urine and sera of the patients.

Bioinformatic tools to predict antigenic epitopes that are specific for the pathogens have been advancing the development of new diagnostics. Many epitopes have been identified by immunoinformatics prediction or immunoassays and the peptides validated by serological assays, such as ELISA [22,23,24,25]. Epitope-based chimeric proteins have been then proposed as recombinant proteins for the development of new COVID-19 vaccines or diagnostics, showing stronger immunogenicity with the possibility to express proteins in a simpler prokaryotic system [13,14]. In our study, N4S11-SC2 showed 83.7% and 91.1% sensitivity when using serum and urine samples, respectively, and 100% specificity for both. Another recombinant chimeric protein of SARS-CoV-2, containing epitopes identified through SPOT synthesis analysis, had shown promise as a serodiagnostic test when using serum samples [19] Taken together, our current study, supported by these additional studies, suggests that polypeptide-based recombinant antigens are promising alternative candidates for the serodiagnosis of SARS-CoV-2. The production of SARS-CoV-2 antibodies typically occurs within 1 to 3 weeks post symptoms onset (PSO) [2]. Thus, one possible reason for our inability to achieve a 100% sensitivity performance, as described for some tests reported in the literature or commercially available, could be the inclusion in our study of samples collected during the early days of PSO. The immune conversion of IgG antibodies in urine and serum samples, with an increase in the IgG levels along the PSO days, were observed for the N4S11-SC2 protein in patients with more than one collection over time, and is herein represented by eight patients in Appendix A. In general, samples from patients that had a negative index on the initial days of collection, from the 21^st^ day onwards, already had an index > 1.1 for serum and urine. This finding corroborates previous studies conducted with the Spike [11] and Nucleocapside proteins [12].

A variety of recombinant antigens for the diagnosis of SARS-CoV-2 are available in the market and described in the literature, although with reports of variable performance, as a consequence of the nature of the antigen used [10,26]. The SARS-CoV-2 N protein can be efficiently expressed with prokaryotic system(s) maintaining good immunoreactivity. However, the S protein has been expressed preferentially in eukaryotic systems, which often generate post-translational modifications of the antigen [10]. Differing from the standard serum-based assay, the prokaryotic expression of the rSARS-CoV-2 S protein was not a barrier to obtain relatively high efficiency for the urine-based ELISA. In this context, the SARS-CoV-2 S protein expressed in the prokaryote system is still not suitable for the detection of anti-SARS-CoV-2 antibodies in sera samples [11]. In this sense, the N4S11-SC2 protein has demonstrated its potential as an alternative recombinant protein expressed in a prokaryotic system, showing the capability to recognize antibodies against N and S proteins of SARS-CoV-2 in serum or urine samples. Given that urine is not yet in practical use, the employment of the N4S11-SC2 protein also presents advantages of expression in a prokaryotic system, as well as its use against serum and urine samples from patients.

Nonetheless, our study has some limitations, such as the absence of a more diverse serological panel, including samples from other variants of SARS-CoV-2 patients and from other cross-reactive diseases. We have not tested our in-house ELISA against samples obtained from patients with respiratory infections caused by other types of coronaviruses, common circulated viruses (influenza, measles and parvovirus, arboviruses and other types of coronavirus, such as HCoVs, that cause only a mild respiratory infection) and individuals vaccinated against influenza. However, we observed no humoral reactivity in serum samples from individuals collected before 2019, sounding that the N4S11-SC2 protein may be specific for SARS-CoV-2 in our in-house ELISA. These same control samples were previously tested using N and S antigens with results of high specificity [11,12]. The patients enrolled in our study had not received any vaccinations at the time of sample collection, so it can be reasonably inferred that the antibodies detected were predominantly produced in response to exposure to the virus.

It is known that the immune response that occurs during vaccination differs from the natural infection. The presence of antibodies generated from the vaccination process was here evaluated using samples from Ad26.COV2.S-vaccinated individuals. Here, we observed that anti-N4S11-SC2 antibodies were not clearly identified using either serum or paired urine samples. The N4S11-SC2 protein exhibited low reactivity against samples collected from vaccinated subjects, particularly when serum samples were used, thus showing a potential to distinguish between those only vaccinated from infected individuals. This may be explained because of the many variable immunological responses found among the population, influenced by age, immunological status, asymptomatic manifestation, doses and type of vaccine taken, which could impact the intensity and durability of antibodies [27]. Naranbhai et al. have compared the immunogenicity elicited by the available vaccines and found that Ad26.COV2.S yielded lower antibody concentrations, which corroborate the results found here [28]. They also found significant variation in antibody concentrations depending on prior infection, vaccine type and vaccine dose and that recipients of Ad26.COV2.S without prior infection had approximately 25-fold lower antibody concentrations than convalescent unvaccinated individuals. Additionally, due to the high infectivity of COVID-19, it is also challenging to differentiate between the extents of infection-induced immunity versus vaccine-induced immunity within a population [29]. However, we cannot discard the possibility of a low binding ability of the S epitopes included in N4S11-SC2. In this sense, a greater number of samples and a good characterization of their variables would be crucial for understanding the presence of N4S11-SC2 antibodies from the vaccine.

Comparing SARS-CoV-2 with six other HCoVs for the N and S protein sequences shows that SARS-CoV-2 and SARS-CoV have the highest sequence identity, 72% (S) and 88% (N), and the other five types are below 50% [30]. A multi-alignment of N4S11-SC2 with other coronaviruses revealed a significant similarity between some of the N4S11-SC2 selected epitopes and SARS-CoV-1, while showing less similarity with MERS-CoV and common cold coronaviruses (Appendix A). However, it seems commonly found in other studies. While there may be some potential cross-reactivity between SARS and MERS, it is important to note that these two coronaviruses do not typically co-occur worldwide. Previous studies revealed that SARS-CoV-1 memory B cell responses tend to be short-lived after infection [31]. The main concern for antigen test cross-reactivity lies with the common cold coronaviruses (HCoV-229E, HCoV-OC43, HCoV-NL63, HCoV-HKU1), however, we found low similarity in the epitopes used here [24].

It is worth noting that the N4S11-SC2 protein was designed based on the Wuhan-Hu-1 isolate sequence (YP_009724390.1) and that antibodies were herein identified in samples collected in 2020/2021, when gamma and previous variants were circulating in Brazil [32]. The use of multiple segments of S and N proteins has been proposed as a way to potentially keep test accuracy as new variants arise [33]. An alignment among the amino acid sequences of all peptides from the N4S11-SC2 chimera and some of the key variants of interest was conducted, revealing few amino acid variations among the analyzed strains (Appendix A). Yet, comprehensive studies on variant effects on the sensitivity of approved antibody tests, where continuous monitoring and assessing kits for prompt variant detection remain crucial, are lacking. Regarding the appearance of new SARS-CoV-2 variants, amino acid mutation may impact the sensitivity of any available test, including our in-house ELISA N4S11-SC2, so future studies are still required to be conduct in order to establish assay sensitivity in patients infected with actual and emerging SARS-CoV-2 variants.

In summary, our data should be considered as providing proof-of-concept of N4S11-SC2 for SARS-CoV-2 diagnosis when using sera and urine samples. They suggest that N4S11-SC2 deserves to be evaluated in large populations in future studies and that its suitability as a diagnostic antigen should also be examined on other platforms, e.g., in rapid point-of-care lateral flow tests.

## Figures and Tables

**Figure 1 viruses-15-01877-f001:**
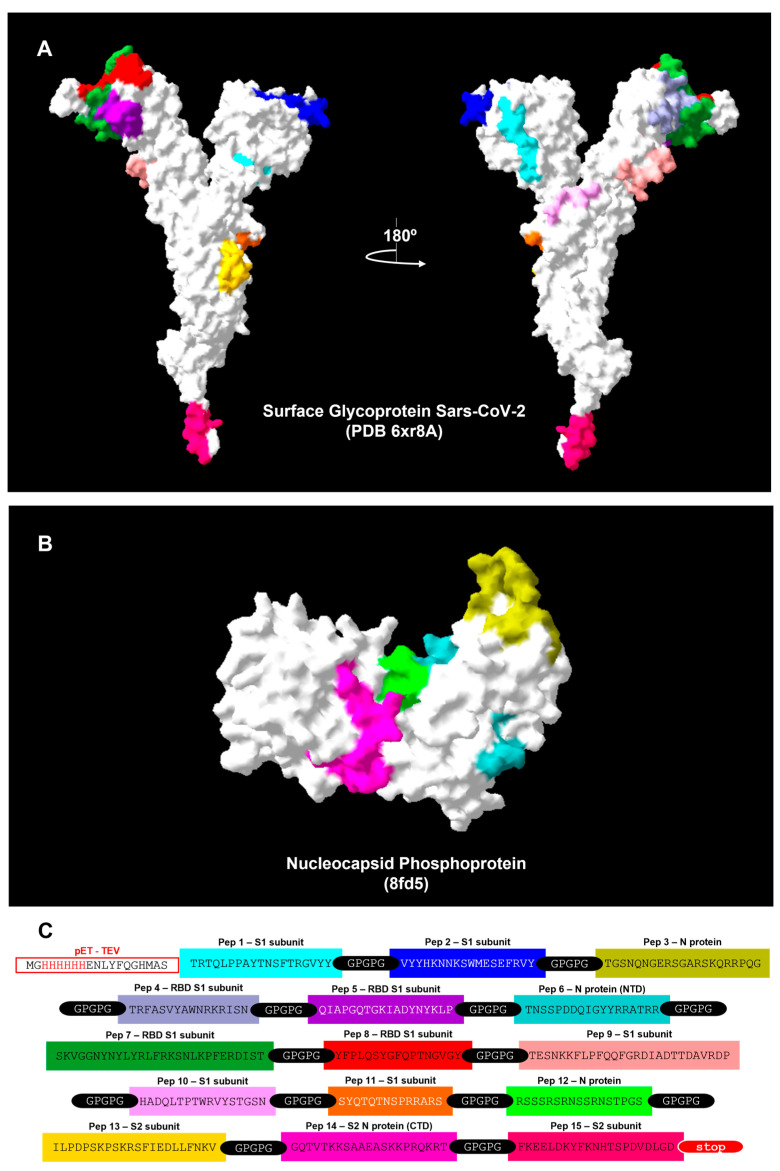
**Construction of the N4S11-SC2 chimeric protein**. (**A**,**B**): Identification of the B-cell epitopes chosen to build the chimeric protein represented on the 3D structure of (**A**) surface glycoprotein (YP_009724390.1) (PDB: 6xr8A) and (**B**) the nucleocapsid phosphoprotein (YP_009724397.2) (PDB: 8FD5). Structures were obtained from PDB and epitopes were selected and identified using SwissPDB-viewer. (**C**): The selected B-cell epitopes were grouped in a linear sequence with the inclusion of –GPGPG- linker residues between each epitope, and the chimeric protein sequence is shown. N4S11-SC2 gene was inserted into pET-TEV expression vector using NheI and NotI restriction enzymes. pET-TEV N-terminal tag containing the initial M and 6X-HIS was maintained into the constructed and a STOP codon was added. Each sequence is represented by color on the 3D structure (**A**,**B**). Peptides 3 and 12 are not represented as the amino acids are not available on the 3D structure.

**Figure 2 viruses-15-01877-f002:**
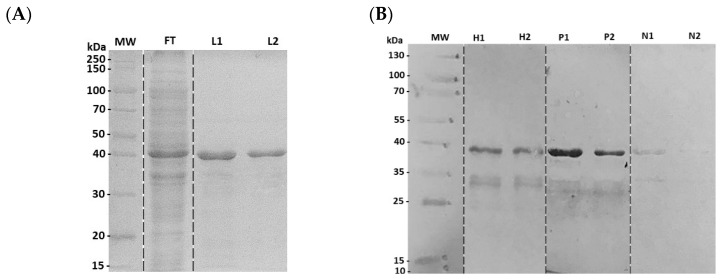
SDS-polyacrylamide gel electrophoresis showing the purification of N4S11-SC2 and immunoblotting of N4S11-SC2 protein against anti-HIS antibody and pooled serum samples from COVID-19 patients and negative control individuals. (**A**) SDS-polyacrylamide gel electrophoresis showing the purification of N4S11-SC2. FT = Flow-Through; L1 = First N4S11-SC2 Elution; L2 = Second N4S11-SC2 Elution; MW = PageRuler™ Unstained Broad Range Protein Ladder. (**B**) H1 and H2: First and second elutions of anti-HIS antibody; P1 and P2: First and second elutions of COVID-19 positive patient’s serum pools; N1 and N2: First and second elutions of negative pre-pandemic individuals’ serum pools; MW = Page Ruler™ Pre-stained Protein Ladder.

**Figure 3 viruses-15-01877-f003:**
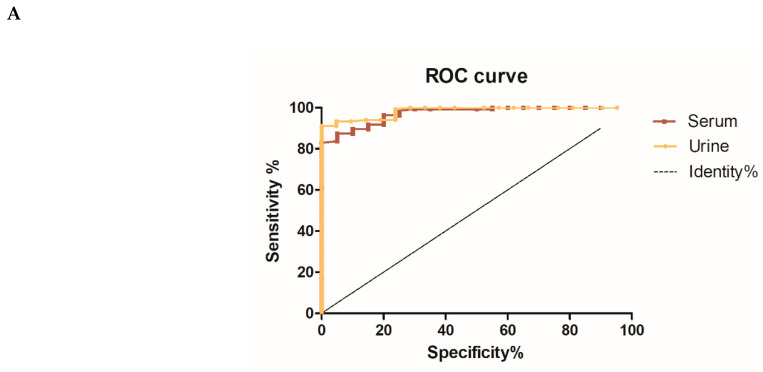
Comparative diagnostic performance of N4S11-SC2 protein with urine and serum samples. (**A**) Receiver Operating Characteristic (ROC) curves were constructed using the individual index (I) value for each sample to obtain sensitivity, specificity and area under the curve values. (**B**) ELISA assays were done using urine and paired serum samples (*n* = 135) from COVID-19 patients with previously positive qRT-PCR. Urine and paired serum samples (*n* = 40) were analyzed from previously vaccinated subjects. Urine and unpaired serum samples from healthy control subjects (*n* = 21 and *n* = 20, respectively) were also used. The mean of each group is shown and the gray band indicates indeterminate values for each sample, while index values below the range (<0.8) are negative and values above (>1.1) are considered positive.

**Figure 4 viruses-15-01877-f004:**
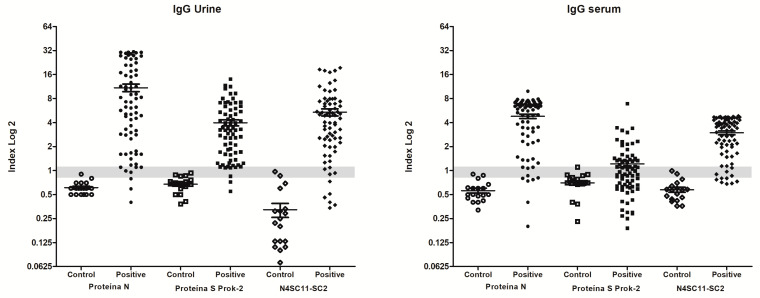
**Comparative diagnostic performance among the recombinant N, S and N4S11-SC2 proteins.** ELISA assays with the N, S and N4S11-SC2 were conducted using the same urine and serum samples from COVID-19 patients with previously positive qRT-PCR above 8 days PSO (*n* = 72) and from healthy control subjects (*n* = 18). The mean of each group is shown and the gray band indicates indeterminate values for each sample, while index values below the range (<0.8) are negative and values above (>1.1) are considered positive.

**Table 1 viruses-15-01877-t001:** Comparative table of the results obtained from the diagnostic test of N4S11-SC2 protein for COVID-19 based on the search for specific IgG antibodies in urine and serum. Samples from symptomatic patients for COVID-19 and with PCR + for SARS-CoV-2, as well as from negative control subjects were used. The individual index (I) value obtained by the Abs/cut-off ratio was used to construct ROC curves. The diagnostic performance of the antigen in relation to the type of sample used was based on the evaluation of sensitivity (95% CI), specificity (95% CI), Area Under the Curve (AUC) and Youden index (J). Legend: J = (Se + Sp) − 1; *n* = samples number, + = positive sample, − = negative sample. Positive and negative predictive values (PPV and NPV, respectively) were calculated on the basis of the index value, excluding the indeterminate value samples (2 urine and 3 serum samples in negative control group, and 8 urine and 16 serum samples in positive group, had an indeterminate index between 0.8 and 1.1) and using the following equations: NPV = true negative/false negative + true negative and PPV = true positive/false positive + true positive. CI: confidence interval.

Sample	AUC	*p*-Value	Cut-Off (Abs)	Se	95%CI	Sp	95%CI	J	PPV	NPV
URINE	0.9841	<0.0001	>0.1900	91.11	84.99% to 95.32%	100.0	83.89% to 100.0%	0.9111	1.000	0.679
SERUM	0.9722	<0.0001	>0.7525	83.70	76.37% to 89.50%	100.0	83.16% to 100.0%	0.7525	1.000	0.607

## Data Availability

All data needed to evaluate the conclusions in the paper are present in the paper and/or the Appendix A.

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
