# Peer review of "B-Cell Epitopes-Based Chimeric Protein from SARS-CoV-2 N and S Proteins Is Recognized by Specific Antibodies in Serum and Urine Samples from Patients"

_viruses, 2023, doi:10.3390/v15091877_

Round 1

Reviewer 1 Report

Overall this is an interesting study demonstrating a useful antigen for measuring antibody responses and showing that antibodies in the urine have sufficient positive and negative predictive  value to be useful in large population screening studies. However there are a number of weaknesses in the description of the experiments, and in extending these results to the current setting.

1.     The description of the informatics is exceptionally weak. We are told that B cell epitopes were identified using IEDB and another website. Yet when one goes to the IEDB website there are a number of different algorithms available, which were used? The parameters listed in the paper do not fully define the parameter choices in the different algorithms. How did the authors determine which of the epitopes to use in the antigen? Were informatic studies performed to determine which peptides are most likely to cross react with known SARS-CoV-2 variants, and not react with non-SARS corona viruses? A supplemental figure shows the alignments, but was the choice of peptides merely done by eyeball or was an algorithm employed?

2.     ELISA results are highly normalized, using the mean of negative samples plus 4.5 SDs as the index. Somewhere in the paper we need to be given the actual OD values that are used to calculate this, and possibly a graph like figure 3 or 4, but plotting actual ODs of the positive and negative samples. This information is necessary to give us an idea of the signal-to-noise ratio.

3.     It appears that all the samples were collected during one period of the pandemic (although we can’t be sure). What variant was dominant in Brazil at that time? Has the antigen been tested in people who have been infected with Omicron-based variants (in which immunoselection has occurred)? Without such data, this antigen and the entire data set may be irrelevant to current needs.

4.     The authors indicate that samples in COVID+ patients were collected at different time3 points (1,3,7, and 14 days). It is important to know the time frame of antibody development, particularly in the urine. Which sample time points were the data in table I, figures 3 and 4, and all those glowing PPV and NPV derived from? At what time point does the urine become positive relative to serum?

There are minor errors in grammar and spelling. Otherwise OK

Author Response

Reviewer 1

Overall this is an interesting study demonstrating a useful antigen for measuring antibody responses and showing that antibodies in the urine have sufficient positive and negative predictive value to be useful in large population screening studies. However, there are a number of weaknesses in the description of the experiments, and in extending these results to the current setting.

We would like to thank the Reviewer for all the valuable considerations. In the revised version of the paper, we have worked aiming to answer the queries raised, with the purpose to make the study suitable to be published in Viruses.

  1. The description of the informatics is exceptionally weak. We are told that B cell epitopes were identified using IEDB and another website. Yet when one goes to the IEDB website there are a number of different algorithms available, which were used? The parameters listed in the paper do not fully define the parameter choices in the different algorithms. How did the authors determine which of the epitopes to use in the antigen? Were informatic studies performed to determine which peptides are most likely to cross react with known SARS-CoV-2 variants, and not react with non-SARS corona viruses? A supplemental figure shows the alignments, but was the choice of peptides merely done by eyeball or was an algorithm employed?

Answer: Thank you for your attention to all of these details, it has surely contributed to our research which will also reflect in future projects. We hope this answer satisfactorily clarifies your question.

The information on lines 143 to 156 was completed to better address the concerns regarding the parameter used in order to give all the important details. The B-Cell epitope prediction tool from the IEDB server was used to identify the most accessible amino acids in the primary structures through the parameter “Antigen Sequence Properties”. The method Emini Surface Accessibility Prediction was chosen with different window sizes of 14,12, and 10. These parameters suggest a score for each residue, and all the amino acids with a score above 1.0 were considered and marked on the sequence. The ABCpred server was also used to predict the B-cell epitopes with window sizes of 14,12, and 10.  All the sequences with a threshold value above 0.85 were considered for this website. To finally select the sequences, overlapping regions were assembled using the Clustal Ômega tool. We used this tool to compare all 6 predictions (from each website and each window) with each other and with the existing literature. For this, we selected articles with different bioinformatics approaches and articles presenting sequences confirmed with in vitro/in vivo tests. After the alignment, the most frequent regions to be suggested by the analysis were observed and aspects like sequence length and position of the sequence in the 3D structure were considered to select the final predicted epitopes. For this last step, we observed the suggested sequences in the 3D structure if they were positioned on the protein surface, and positioned in a way to facilitate biding. This was also used as some selection criteria, to choose the final epitopes. This method of selection is used by the group based on the experience and knowledge of the researchers and is part of our know-how. It has also been previously used to predict epitopes from different proteins in other projects, some of which have been confirmed by in vitro and in vivo analysis.

Regarding cross-reaction, a figure containing the alignment of the chosen sequences with different variants of the Sars-CoV-2 was added to the file (Supplementary Figure 3). The regions were compared with the variants Alpha, Beta, Gama, Delta, and Omicron, where we observed very few alterations, suggesting that the chosen regions are highly conserved between variants. Nonetheless, it is of great importance to test the N4S11-SC2 protein against samples of patients infected with different existing variants as well as other ones that may emerge to ensure its effectiveness and accuracy, and this may be considerate a limitation of our study, although we may keep in mind that this is still an in house ELISA study with potential to become an innovation.

We have also included a discussion about this concern in the paper:

“It is worth noting that the N4S11-SC2 protein was designed based on the Wu-han-Hu-1 isolate sequence (YP_009724390.1) and that antibodies were herein identi-fied in samples collected in 2020/2021, when gamma and previous variants were cir-culating in Brazil (Alcantara et al 2022). The use of multiple segments of S and N proteins have been proposed as a way to potentially keep test accuracy as new variants arise (Thakur et al., 2022). An alignment among the amino acid sequences of all peptides from the N4S11-SC2 chimera and some of the key variants of interest was conducted, revealing few amino acid variations among the analyzed strains (Fig. S3). Yet, comprehensive studies are lacking on variant effects on approved antibody tests' sensitivity, where continuous monitoring and assessing kits for prompt variant detection remaining crucial. Regarding the appearance of new SARS-CoV-2 variants, amino acid mutation may impact the sensitivity of any available test, including our in house ELISA N4S11-SC2, so future studies are still required to be conduct in order to establish assay sensitivity in patients infected with actual and emerging SARS-CoV-2 variants.”

Regarding the possible reaction with non-Sars coronaviruses, we have conducted a multialigment with some other HCoVs S and N sequences as shown in Supplementary Figure 2. We also discussed this point in the main text. However, it would be of great interest to test the N4S11-SC2 protein against different coronaviruses in future analyses to confirm this theory and further consolidate our results.

“Comparing SARS-CoV-2 with six other HCoVs for the N and S protein sequences shows that SARS-CoV-2 and SARS-CoV have the highest sequence identity, 72% (S) and 88% (N), and the other five types are below 50% (Zhang et al, 2022). A mul-ti-alignment of N4S11-SC2 with other coronaviruses revealed a significant similarity between some of N4S11-SC2 selected epitopes and SARS-CoV-1, while showing less similarity to MERS-CoV and common cold coronaviruses (Fig. S2). However, it seems a commonly found in other studies. While there may be some potential cross-reactivity between SARS and MERS, it is important to note that these two coronaviruses do not typically co-occur worldwide. Previous studies revealed that SARS-CoV-1 memory B cell responses tend to be short-lived after infection (Channappanavar et al, 2014). The main concern for antigen test cross-reactivity lies with the common cold coronaviruses (HCoV-229E, HCoV-OC43, HCoV-NL63, HCoV-HKU1) (Rodrigues-da-Silva et al, 2023), however, we found low similarity in the epitopes here used.”

  1. ELISA results are highly normalized, using the mean of negative samples plus 4.5 SDs as the index. Somewhere in the paper we need to be given the actual OD values that are used to calculate this, and possibly a graph like figure 3 or 4, but plotting actual ODs of the positive and negative samples. This information is necessary to give us an idea of the signal-to-noise ratio.

Answer: Thank you for pointing out this gap in our manuscript. In fact, the value of 4.5 SDs would be a really high normalization number. We do double checked this number and found typo. We fixed the number in the text to the correct values used to calculate the index: negative samples plus 2.5 SDs for urine and 2.3 SDs for serum.

“The cut-off values were determined as the mean plus 2.5 and 2.3 times the standard deviation of negative samples for urine and serum assay, respectively.”

  1. It appears that all the samples were collected during one period of the pandemic (although we can’t be sure). What variant was dominant in Brazil at that time? Has the antigen been tested in people who have been infected with Omicron-based variants (in which immunoselection has occurred)? Without such data, this antigen and the entire data set may be irrelevant to current needs.

Answer: We also agree that this is a very curious point, but unfortunately at this moment, we cannot extend our work to solve this information because the samples that we have available were collected in 2020/2021. However, we keep it in mind and we hope to be able to solve this issue in future studies. This is a limitation of our work at this moment, and by now, we cannot assume that antibodies will be detected by different strains of the virus. This has been considering a limitation of any immunodiagnostic available where conduction of revalidation has been suggested for a confident result as variants arise.

We added this sentence to our manuscript:

It is worth noting that the N4S11-SC2 protein was designed based on the Wuhan-Hu-1 isolate sequence (YP_009724390.1) and that antibodies were herein identified in samples collected in 2020/2021, when gamma and previous variants were circulating in Brazil (Alcantara et al 2022). The use of multiple segments of S and N proteins have been proposed as a way to potentially keep test accuracy as new variants arise (Thakur et al., 2022). An alignment among the amino acid sequences of all peptides from the N4S11-SC2 chimera and some of the key variants of interest was conducted, revealing few variable variations among the analyzed strains (Fig. S3). Yet, comprehensive studies are lacking on variant effects on approved antibody tests' sensitivity, where continuous monitoring and assessing kits for prompt variant detection remaining crucial. Regarding the appearance of new SARS-CoV-2 variants, amino acid mutation may impact the sensitivity of any available test, including our in house ELISA N4S11-SC2, so future studies are required to be conduct in order to establish assay sensitivity in patients infected with present and future SARS-CoV-2 variants.

  1. The authors indicate that samples in COVID+ patients were collected at different time3 points (1,3,7, and 14 days). It is important to know the time frame of antibody development, particularly in the urine. Which sample time points were the data in table I, figures 3 and 4, and all those glowing PPV and NPV derived from? At what time point does the urine become positive relative to serum?

Answer: This is a very important point, specially because of the time frame for the immunological system to start generating antibodies. We had decided to not include this information in this paper as it was previously established and well discussed in our previous studies conducted with the N and S proteins (Ramos et al 2023 and Ludolf et al 2022). However, reading your comments we do agree that it is very important to not lose this information also in the present study.

In preparing the table and Figure 3, we used all available samples, which contain samples starting from the 2nd post symptoms onset (n=135).

On the other hand, for Figure 4, the samples used for the comparison between the three proteins were above 8 days PSO (n=72) in order to keep them paired. Again, the results of N and S proteins are reproducible, as evidenced in previous works, such as Ludolf, 2022 and Ramos, 2023, respectively.

We have included the information in the text:

Immune conversion of IgG antibodies in urine and serum samples, with an increase in IgG levels along the PSO days were observed for the N4S11-SC2 protein in patients with more than one collection over time, and is herein represented by eight patients in Supplementary Fig 4. In general, samples from patients that had a negative index on the initial days of collection, from the 21st day onwards, already had an index >1.1 for serum and urine. This found corroborate with previous studies conducted with the Spike (Ramos et al., 2023) and Nucleocapside (Ludolf et al., 2022) proteins.

Supplementary figure 3. Dynamics of IgG antibody conversion in patient urine and serum samples. Figures show the IgG levels specific to the N4S11-SC2 for eight patients, with longitudinal collection on different days post-symptom onset. The plotted index values (I) were related to the absorbance ratio on the cut-off.

Another point addressed, the values of PPV and NPV are very good data, but it is worth mentioning that the data of the participating individuals were carefully collected, providing a reliable classification of the samples in their respective groups. In addition, these analyzes were calculated based on the index values obtained for the samples, excluding the samples with indeterminate values.

We have included this information in the table legend:

(2 urine and 3 serum samples in negative control group, and 8 urine and 16 serum samples in positive group, had an indeterminate index between 0.8 and 1.1.),

Reviewer 2 Report

Here the authors designed a new antigen combining 15 different epitopes from SARS2 S and N protein. Further experiments showed that this new antigen has well-expression. The authors went on to demonstrate that binding ability against serum and urine samples from COVID-19 patients.

This is a well-designed and well-written study that shows a potential replaceable method to detect SARS2 although some details were not clear enough and its S protein binding ability is doubtful. I thought the data was basicely rigorous and several questions remain.

1.     Are the chosen epitopes conservative among different sars2 mutant strains? The author can perform sequence alignment and mark the epitopes in Fig.1

2.     Fig.1C showed the combination of 15 peptides. I’m just curious what determines their order? Is it working if peptide 15 is adjusted to the beginning?

3.     In Fig.3, N4S11-SC2 has poor response against vaccinated serum.  Although N4S11-SC2 could recognize IgG serum better than WT S, but it was weaker if compared with WT N protein in Fig.4. Does it imply that the chosen S epitopes in N4S11-SC2 have low binding ability?Please explain and discuss it.

Author Response

Reviewer 2

Here the authors designed a new antigen combining 15 different epitopes from SARS2 S and N protein. Further experiments showed that this new antigen has well-expression. The authors went on to demonstrate that binding ability against serum and urine samples from COVID-19 patients.

This is a well-designed and well-written study that shows a potential replaceable method to detect SARS2 although some details were not clear enough and its S protein binding ability is doubtful. I thought the data was basicely rigorous and several questions remain.

Answer: Firstly, we would like to thank the Reviewer 2 for the positive evaluation of our article. In the revised version of the manuscript, we have worked thoroughly to answer all queries raised, in order to improve its quality, aiming to make it more suitable to be published in this renowned scientific journal.

  1. Are the chosen epitopes conservative among different sars2 mutant strains? The author can perform sequence alignment and mark the epitopes in Fig.1

Thank you for the suggestion of analyzing the potentially sequence mutation in the selected epitopes among the Sars2 mutant strains. We have conducted this alignment and included a new figure (3) as supplementary material. We have also included some discussion regarding this information in the main text:

It is worth noting that the N4S11-SC2 protein was designed based on the Wuhan-Hu-1 isolate sequence (YP_009724390.1) and that antibodies were herein identified in samples collected in 2020/2021, when gamma and previous variants were circulating in Brazil (Alcantara et al 2022). The use of multiple segments of S and N proteins have been proposed as a way to potentially keep test accuracy as new variants arise (Thakur et al., 2022). An alignment among the amino acid sequences of all peptides from the N4S11-SC2 chimera and some of the key variants of interest was conducted, revealing few variable variations among the analyzed strains (Fig. S3). Yet, comprehensive studies are lacking on variant effects on approved antibody tests' sensitivity, where continuous monitoring and assessing kits for prompt variant detection remaining crucial. Regarding the appearance of new SARS-CoV-2 variants, amino acid mutation may impact the sensitivity of any available test, including our in house ELISA N4S11-SC2, so future studies are required to be conduct in order to establish assay sensitivity in patients infected with present and future SARS-CoV-2 variants.

  1. 1C showed the combination of 15 peptides. I’m just curious what determines their order? Is it working if peptide 15 is adjusted to the beginning?

Answer: Thank you for the question. We decided to include this information in the paper thinking that other readers might have the same curiosity as you. Explaining: The final sequences from both proteins were distributed in the new protein in order to preserve sequences in which appear in the original proteins. For example, sequences that were close to the N-terminal portion in the original protein were kept in that region, and sequences close to the C-terminal were arranged to be as close as possible in the new protein. After the selection and assembly of the N4S11-SC2 protein, its sequence was again run through the websites to ensure the regions would still be suggested as antigenic. The sequence of amino acids is directly linked to the spatial configuration and therefore to the functionality of the protein. Other considered aspects were the solubility and stability of the new protein, which were confirmed by Expasy tools. The sequence of amino acids can also interfere with these parameters, so is possible that a different order would not meet the selected criteria to fulfill the protein’s purpose. Likewise, it is possible that changing the arrangements would not change the final result, as long as crucial parameters like solubility, stability and conformational structure were kept.

Information was added to enlighten these matters.

The arrangement of the peptides in the chimeric protein was distributed in a position as similar as possible to the original proteins, preserving the order in which they appear. The Expasy tool was used for the physical-chemical characterization of the multi-epitope chimeric protein, which was called N4S11-SC2 (Fig.1C). In this step, the new protein was analyzed again by ABCPred to confirm if the peptides would still be suggested as antigenic regions.

  1. In Fig.3, N4S11-SC2 has poor response against vaccinated serum.  Although N4S11-SC2 could recognize IgG serum better than WT S, but it was weaker if compared with WT N protein in Fig.4. Does it imply that the chosen S epitopes in N4S11-SC2 have low binding ability? Please explain and discuss it.

Answer: We also agree that this is a very curious point, however at this moment, we can just hypnotize the results we have found and they are discussed in the text.

We also included the text below:

Naranbhai et al 2022 have compared the immunogenicity elicited by the available vaccines and found that Ad26.COV2.S yielded lower antibody concentrations, which corroborate with the results herein found. They also found significant variation in antibody concentrations depending on prior infection, vaccine type, and vaccine dose and that recipients of Ad26.COV2.S without prior infection had an approximately 25-fold lower antibody concentrations than convalescent unvaccinated individuals.

However, we cannot discard the possibility of a low binding ability of the S epitopes included in N4S11-SC2.

Round 2

Reviewer 1 Report

The authors have done an excellent job in addressing my concerns and revising the manuscript

Minor concerns